# Identifying the Structural Adaptations that Drive the Mechanical Load-Induced Growth of Skeletal Muscle: A Scoping Review

**DOI:** 10.3390/cells9071658

**Published:** 2020-07-09

**Authors:** Kent W. Jorgenson, Stuart M. Phillips, Troy A. Hornberger

**Affiliations:** 1School of Veterinary Medicine and the Department of Comparative Biosciences, University of Wisconsin, Madison, WI 53706, USA; kwjorgenson@wisc.edu; 2Department of Kinesiology, McMaster University, Hamilton, ON L8S 4K1, Canada; phillis@mcmaster.ca

**Keywords:** fascicle, myofiber, myofibril, sarcomere, hypertrophy, hyperplasia, splitting, radial growth, longitudinal growth, exercise

## Abstract

The maintenance of skeletal muscle mass plays a critical role in health and quality of life. One of the most potent regulators of skeletal muscle mass is mechanical loading, and numerous studies have led to a reasonably clear understanding of the macroscopic and microscopic changes that occur when the mechanical environment is altered. For instance, an increase in mechanical loading induces a growth response that is mediated, at least in part, by an increase in the cross-sectional area of the myofibers (i.e., myofiber hypertrophy). However, very little is known about the ultrastructural adaptations that drive this response. Even the most basic questions, such as whether mechanical load-induced myofiber hypertrophy is mediated by an increase in the size of the pre-existing myofibrils and/or an increase in the number myofibrils, have not been resolved. In this review, we thoroughly summarize what is currently known about the macroscopic, microscopic and ultrastructural changes that drive mechanical load-induced growth and highlight the critical gaps in knowledge that need to be filled.

## 1. Introduction

Skeletal muscle comprises approximately 40% of body mass and plays a critical role in posture, breathing, motion, and metabolic regulation [1]. As we age, the occurrence of age-related diseases, such as the loss of muscle mass (i.e., sarcopenia), are expected to become more prevalent [2]. For instance, between the ages of 25–80 years, the average individual will lose approximately 25% of their muscle mass [3,4]. This age-associated loss of muscle mass leads to an increased risk of fall-related injury, institutionalization, loss of independence, and disease [5,6,7]. Indeed, in the United States alone, the healthcare costs for muscle wasting related illnesses were estimated to be $18.5 billion in 2000 [8]. Based on this figure, reducing the rate of muscle wasting related diseases by even 10% could save a striking $1.1 billion in annual healthcare costs. The number of people over the age of 60 is expected to double by 2050, and thus, the costs associated with sarcopenia will only continue to increase [9]. Accordingly, the development of therapies that can restore, maintain, and/or increase muscle mass will be of great clinical and fiscal significance. However, to develop such therapies, we will first need to establish a comprehensive understanding of the mechanisms that regulate the size of this vital tissue.

Mechanical load-induced signals are one of the most widely recognized regulators of skeletal muscle mass. Indeed, historical evidence suggests that the growth-promoting effects of mechanical loading has been recognized since at least the 7th century BC [10]. During the last century, a variety of human and animal models have been used to further establish this point. For instance, in humans, resistance exercise is the most commonly used model of mechanical load-induced growth and it typically induces a 5–20% increase in skeletal muscle volume/mass within 8–16 weeks [11,12,13,14,15,16,17]. Similar changes in muscle mass have also been observed in animal models that are intended to mimic human resistance exercise [18,19,20]. Furthermore, animal models that use extreme forms of mechanical loading, such as synergist ablation, can promote a doubling of muscle mass within as little as 2 weeks [21,22,23]. Collectively, these models have provided extensive insight into the macroscopic and microscopic changes that contribute to the mechanical load-induced growth response, but surprisingly, the ultrastructural changes that drive these changes remain poorly understood. In this review, we will thoroughly summarize what is currently known about the structural adaptations that drive mechanical load-induced growth and highlight the critical gaps in knowledge that need to be filled.

## 2. Overview of Skeletal Muscle Structure

Before considering the structural changes that drive mechanical load-induced growth, we want to ensure that the reader appreciates the basic structural design of skeletal muscle. One of the easiest ways to appreciate this design is to consider skeletal muscle as a hierarchy of contractile machinery that is visible at the macroscopic level (viewable without magnification), followed by the microscopic level (viewable with standard microscopy), and finally the ultrastructural level (viewable with high resolution microscopy). Below we will provide a brief overview of the primary components that are found at each of these levels. For excellent illustrations and more comprehensive discussions on this topic, the reader is referred to the following reviews [24,25,26].

At the macroscopic level, it can be noted that skeletal muscles are connected to bones via tendinous attachments and enact their contractile function by providing movement and articulation of the skeletal system. Moreover, as illustrated in Figure 1, skeletal muscles are surrounded by an outer layer of connective tissue called the epimysium, and underneath the epimysium are bundles of myofibers (i.e., fascicles) that are surrounded by another layer of connective tissue called the perimysium [24]. In most skeletal muscles, the fascicles, and their associated myofibers, are not directly aligned with the longitudinal axis of the muscle, but instead are offset at an angle called the pennation angle (Figure 2) [27].

At the microscopic level, a cross-section of skeletal muscle will reveal the presence of individual myofibers (Figure 1A,B). The myofibers are multinucleated cells that are encased by a layer of connective tissue called the endomysium, and they are surrounded by interstitial cells such as fibroblasts, immune cells, pericytes and fibro-adipogenic progenitors (Figure 1A–C) [29,30]. Furthermore, another important class of cells, called satellite cells, resides between the endomysium and the plasma membrane of the myofibers (i.e., the sarcolemma) [31]. The endomysium is physically coupled to the sarcolemma, and everything the resides underneath the sarcolemma is typically referred to as the sarcoplasm.

The gelatinous sarcoplasm contains the primary ultrastructural elements of the myofiber, and as illustrated in Figure 1, an examination at the ultrastructural level reveals that ≈80% of the sarcoplasm is filled with an in-parallel array of rod-like structures called myofibrils [32,33,34,35,36]. The myofibrils are composed of a long in-series array of force-generating elements called sarcomeres and are surrounded by a mitochondrial reticulum and a membranous structure called the sarcoplasmic reticulum [37,38]. The sarcomeres within the myofibrils enact their function through the active sliding of thick and thin myofilaments, and in a longitudinal view, it can be seen that the sarcomeres consist of regions called the Z-disc, the I-band which contains the thin (actin) myofilaments, and the A-band which contains the thick (myosin) myofilaments [39,40]. It should also be noted that within a given species, the optimal/resting length of sarcomeres (2.0–2.5 μm) is highly conserved, and alterations in this length can profoundly influence force production [41,42,43].

When considering the structure of skeletal muscle, it is also essential to recognize that all myofibers are not created equal. For instance, some types of myofibers are heavily reliant on oxidative metabolism, exhibit a slow contractile speed and are resistant to fatigue. In contrast, other types of myofibers rely on anaerobic glycolytic metabolism, exhibit a fast contractile speed and rapidly fatigue when stimulated to contract [40]. Different fiber types are typically grouped according to the predominant isoform of the myosin heavy chain that they express, and these isoforms include Type I (slow oxidative), Type IIA (fast oxidative), and Type IIB (fast glycolytic) fibers [44]. It is also important to point out that humans do not express the Type IIB myosin isoform, but instead express a very similar (yet slightly slower) Type IIX myosin isoform [45]. However, since this differentiation was only solidified in 1990′s [46,47,48], some older studies with human subjects used the Type IIB classification, while other studies have grouped Type IIB and Type IIX fibers together as a similar fiber type [49]. This use of IIB and IIX myosin labeling has led to some confusion when comparing earlier muscle growth studies to current studies, so it is important to keep this distinction in mind when relating fiber type-specific adaptations across the current body of literature.

## 3. Mechanical Load-Induced Growth of Skeletal Muscle at the Macroscopic Level

### 3.1. Whole Muscle

At the whole muscle level, mechanical load-induced growth can be mediated by an increase in the length and/or an increase in the cross-sectional area (CSA) of the muscle. Growth resulting from an increase in length is referred to as longitudinal growth, and it can occur in response to a variety of different perturbations. For instance, during development, the length of muscles can more than double from birth to the termination of bone growth [50,51,52]. Longitudinal growth can also be induced in adults by placing muscles in a chronically stretched state [53]. As a case in point, it has been shown that immobilizing a rat lower hindlimb in a fully dorsiflexed position can lead to a >20% increase in the length of the soleus muscle [54]. Likewise, limb-lengthening procedures can lead to a >20% increase in muscle length [55,56]. Indeed, even some of the more extreme models of mechanical load-induced growth can lead to an increase in muscle length [23,57]. For example, surgical removal of the gastrocnemius and soleus muscles (i.e., synergist ablation) is a commonly used rodent model for stimulating mechanical load-induced growth, and it has been reported that this can lead to a 13% increase length of the plantaris muscle [57,58]. Thus, it is clear that both adolescent and adult skeletal muscles can undergo longitudinal growth.

Although skeletal muscle is capable of undergoing longitudinal growth, most models of mechanical loading do not lead to notable alterations in whole muscle length [18,59]. Instead, mechanical load-induced growth is usually driven by an increase in the CSA of the muscle (also known as radial growth). For example, in humans, 8-16 weeks of resistance exercise will generally produce a 5–30% increase in whole muscle CSA but no change in muscle length [16,59,60,61,62,63,64,65,66]. Interestingly, the magnitude of increase in CSA is often greater than the increase that is observed for muscle volume/mass. An excellent example of this paradox was reported by Roman et al. (1993), whom reported that 12 weeks of resistance exercise led to a 14% increase in the volume of the elbow flexors, but the CSA at the mid-belly increased by 23% [60]. Importantly, however, the magnitude of the increase is CSA got progressively smaller towards the proximal and distal ends of the muscle, which explained why the muscle volume only increased by 14%. Simply put, the study by Roman et al. (1993) demonstrated that the radial growth response was not evenly distributed along the length of the muscle. Indeed, regional differences in the magnitude of radial growth have been reported in several animal and human-based studies, and in our opinion, this phenomenon represents an often overlooked aspect of the mechanical load-induced growth response [13,59,64,67,68,69,70,71,72].

### 3.2. Muscle Fascicles

Previous studies have shown that the initial mechanical load-induced increase in whole muscle CSA can be attributed, at least in part, to edema. However, the long-term changes are primarily caused by an expansion of the contractile elements [63,73,74]. For instance, using a rat model of resistance exercise, we have shown that 8 weeks of training resulted in a 24% increase in the CSA of the flexor hallucis longus muscle, and this was matched by a proportionate increase in total myofibrillar protein content and peak tetanic force production [18]. Thus, if the mechanical load-induced increase in whole muscle CSA was driven by an expansion of the contractile elements, then the increase should be reflective of the changes that occurred at the preceding level within the hierarchy of the contractile machinery (i.e., myofilaments → myofibrils → myofibers → fascicles → whole muscle).

Based on the aforementioned point, mechanical load-induced changes in whole muscle CSA should be driven by changes that happen at the level of the muscle fascicles, and there are effectively two predominant ways in which this is thought to occur: (1) longitudinal growth of the fascicles or (2) radial growth of the fascicles. It is also possible that mechanical loading could lead to an increase in the number of fascicles per muscle, but we are not aware of any studies that have attempted to answer this technically difficult question.

Upon first consideration, it can be challenging to appreciate how both longitudinal and radial growth of fascicles can lead to an increase in whole muscle CSA. Thus, to visualize these points, we have taken advantage of a geometric model that can be used to predict changes in the architectural properties of skeletal muscle [75,76]. Specifically, as shown in Figure 3, we used this model to illustrate how changes in either fascicle length (Lf), or fascicle diameter (Df), could produce a 30% increase in whole muscle CSA (the upper end of what is typically observed in humans after 8–16 weeks of resistance exercise). For simplicity, our model considers a hypothetical muscle that is composed of 50 fascicles aligned in parallel. Moreover, in the control (starting) state, the fascicle length to muscle length (Lm) ratio is 0.25, and the fascicles are offset at a pennation angle of 16° (similar to the properties of the vastus lateralis muscle in humans [77]). Based on these parameters, if the 30% increase in CSA was purely due to longitudinal growth of the fascicles, then fascicle length would have to increase by 11%, and the pennation angle would remain unaltered (Figure 3B). On the other hand, if the 30% increase in CSA was due exclusively to an increase in radial growth of the fascicles, then the fascicle diameter would have to increase by 14%, and this would result in a concomitant 15% increase in pennation angle (from 16°→18.4°) (Figure 3C). It is also worth noting that in the example of pure longitudinal growth, the number of fascicles visible in a cross-section of the mid-belly of the muscle would also increase by 30%, and could easily lead one to mistakenly conclude that the increase in whole muscle cross-sectional area was driven by new fascicle formation.

Having illustrated how radial and longitudinal growth of fascicles can lead to an increase in whole muscle CSA, we will now consider the studies that have tested whether these types of adaptations occur. Specifically, we will first consider the studies that have examined whether mechanical load-induced alterations in whole muscle CSA are associated with changes in fascicle length, and fortunately, this has been a subject of extensive investigation [13,59,62,65,78,79,80,81,82,83,84,85]. For instance, Ema et al. (2016) recently compiled data from 38 different studies that addressed this topic and found that a significant positive relationship existed between the exercise-induced increases in muscle size and fascicle length [86]. Nonetheless, some of the studies that reported an increase in muscle size did not observe an increase in fascicle length [59,83,84,85], and there are even examples in which small but significant declines in fascicle length have been reported [78]. However, when Ema et al. (2016) compared the magnitude of change across all studies which showed a significant alteration in fascicle length versus those which did not, the average values from these studies still showed a 12.4% versus 7.7% increase in fascicle length, respectively [86]. Thus, there is a high level of support for the notion that mechanical load-induced alterations in whole muscle CSA can be driven, at least in part, by an increase in fascicle length.

As mentioned above, the radial growth of fascicles could also lead to an increase in whole muscle CSA. Importantly, as detailed by the work of Maxwell et al. (1974), and as illustrated in Figure 3C, a direct relationship exists between fascicle diameter and the pennation angle of the fascicles. Specifically, if the length of the muscle, the length of the fascicles, and the number of fascicles is held constant, then an increase in fascicle diameter will lead to an increase in the pennation angle. Hence, it is not surprising that most studies reporting a significant resistance exercise-induced increase in muscle size, but no change in fascicle length, instead find a significant increase in the pennation angle [59,83,84,85]. Indeed, just as with changes in fascicle length, Ema et al. (2016) determined that a significant positive relationship exists between the resistance exercise-induced increase in muscle size and pennation angle. On average, the studies that reported a significant change in pennation angle showed a 13.5% increase, while those that did not detect a significant change still found an average increase of 7.7% [86]. Accordingly, just as with changes in fascicle length, there is a high level of support for the notion that mechanical load-induced alterations in whole muscle CSA can be driven by an increase in fascicle diameter/pennation angle. Indeed, a collective view of the literature suggests that mechanical loading can lead to both longitudinal and radial growth of fascicles, and the exact contribution of these components is probably determined by a variety of different factors, such as the type of mechanical loads that are placed on the muscle (e.g., concentric vs. eccentric contractions) and the architectural properties of the muscle that is being considered (e.g., fusiform, unipennate, bipennate, etc.) [86,87,88,89].

## 4. Mechanical Load-Induced Growth of Skeletal Muscle at the Microscopic Level

As previously noted, mechanical load-induced alterations at each level of the skeletal muscle structure should be reflective of the changes that occurred at the preceding level within the hierarchy of the contractile machinery. Thus, having established that mechanical loading can lead to both longitudinal and/or radial growth of the fascicles, we will now consider how these changes can be mediated by alterations at the level of the myofibers.

### 4.1. Longitudinal Growth of Fascicles

Fascicles are composed of bundles of myofibers, and the myofibers can either run the entire length of the fascicle, or only part of the length of the fascicle and exhibit an intrafascicular termination [90,91,92]. For fascicles that are composed of myofibers that run the entire length of the fascicle, longitudinal growth of the fascicle would be exclusively dependent on the longitudinal growth of the individual myofibers. Alternatively, longitudinal growth of the fascicles with myofibers that exhibit intrafascicular terminations could result from longitudinal growth of the myofibers and/or the addition of new myofibers in-series. Although we are not aware of any studies that have addressed whether mechanical loading can lead to the formation of new myofibers in-series, a consistent body of literature has shown that myofibers are capable of undergoing longitudinal growth [57,72,93,94,95]. For instance, Alway et al. (1989) subjected the anterior latissimus dorsi (ALD) muscle of quails to chronic mechanical loading by securing a weight (10% of body mass) to one of the wings. In response to this perturbation, the mass of the ALD increased by 182%, which was associated with a 24% increase in the average length of the myofibers [72]. Similarly, Roy et al. (1982) demonstrated that in rats, chronic mechanical loading of the plantaris via synergist ablation resulted in doubling of its mass and a concomitant 19% increase in the myofiber to muscle length ratio [94]. Based on these, and related studies, it is clear that extreme models of mechanical loading can induce longitudinal growth of the myofibers.

Although a compelling body of evidence indicates that extreme models of mechanical loading can promote longitudinal growth of myofibers, only a handful of studies have directly addressed this topic within the confines of more physiologically relevant models. For instance, it has been shown that the eccentric contractions induced by downhill walking [96], and downhill running [97], can lead to an increase in the number of sarcomeres per myofiber; however, neither of these studies reported measurements of myofiber length. Indeed, we could only find one study that reported measurements of myofiber length within the context of a physiologically relevant model of mechanical load-induced growth [18]. In this case, rats were subjected to 8 weeks of resistance exercise which led to a 24% increase in whole muscle CSA, but myofiber length was not altered. Importantly; however, this study did not indicate whether the increase in muscle CSA was mediated by longitudinal vs. radial growth of the fascicles, and hence, it is difficult to extrapolate any meaningful insights from the data.

Given the paucity of data on this topic, we believe that it is worthwhile to mention unpublished results that we recently obtained from mice that had their plantaris muscles subjected to 16 days of myotenectomy (a much milder form of synergist ablation [98]). Specifically, we determined that myotenectomy led to 72% increase in the mass of the plantaris along with an 8% increase in length of the myofibers (*p* < 0.01). Likewise, Goh et al. (2019) recently described a high intensity interval training (HIIT) for mice that leads to a 17% increase in the mass of the extensor digitorum longus muscle [99], and this was associated with a 9% increase in the length of the myofibers (*p* < 0.05, personal communication from Dr Doug Millay). Thus, it appears that even physiologically relevant models of mechanical load-induced growth can induce longitudinal growth of myofibers; however, additional studies on this topic will need to be published before a clear consensus can be reached.

### 4.2. Radial Growth of Fascicles

As illustrated in Figure 4, radial growth of fascicles could result from an increase in the CSA of the existing myofibers (i.e., myofiber hypertrophy, Figure 4A) and/or an increase in the number of myofibers per cross-section (from myofiber splitting and/or hyperplasia, Figure 4B). These concepts have been widely studied within the context of mechanical load-induced growth, and in the following sections we will summarize the body of literature that exists on these topics. Before moving into these sections, we also want to point out that the radial growth of fascicles could result from the longitudinal growth of myofibers with intrafascicular terminations (Figure 4C) [91]. However, as mentioned above, very few studies have examined whether physiologically relevant models of mechanical loading can induce longitudinal growth of myofibers. Accordingly, this mechanism will not be subjected to further discussion.

#### 4.2.1. Myofiber Hypertrophy

Radial growth of myofibers leads to an increase in the CSA, and such a change is typically referred to as myofiber hypertrophy. Myofiber hypertrophy is, by far, the longest-standing and most widely acknowledged contributor to the mechanical load-induced growth of skeletal muscle. Indeed, the ability of mechanical loads to induce myofiber hypertrophy has been recognized since the late 1800′s [100]. As summarized by Huan et al. [101], most of the early research on this topic used animals such as dogs [100], cats [102], mice [103], rats [104], hamsters [35], and birds [105]. Some of these animal-based studies employed rather extreme forms of chronic mechanical loading (e.g., synergist ablation, wing-weighting, etc.), whereas others used interventions that were intended to mimic human resistance exercise. A notable example was described by Goldspink (1964) in which young mice were trained to pull on a weighted cord so that they could gain access to their food, and it was determined that 25 days of this training resulted in a ≈30% increase in the CSA of myofibers within the biceps brachii [103]. Another classic example involves the model described by Gonyea and Ericson (1976) [102]. In this model, cats were operantly conditioned to move a weighted bar with their paw in exchange for a food reward, and it was found that the CSA of the myofibers within the flexor carpi radialis increased by 21–32% after 41 weeks of this type of training [102]. The magnitude of change in myofiber CSA observed in the above examples is similar to the 10–35% that is typically observed in humans after 8-16 weeks of resistance exercise [16,17,44,60,66,101,106,107,108]. However, this magnitude of change pales in comparison to what has been observed with some of the more extreme models of mechanical loading. For instance, Antonio and Gonyea (1993) observed an astonishing 142% increase in CSA of the myofibers of the ALD muscle after just 16 days of wing-weighting [109]. Simply stated, an extremely high level of evidence supports the notion that mechanical loading can induce myofiber hypertrophy and the capacity for this type of growth appear to be quite large.

#### 4.2.2. Myofiber Splitting

As recently reviewed by Murach et al. (2019), split myofibers are characterized by the presence of “branching”, “fragmentation”, or “splitting” along the length of the myofiber [110]. Split myofibers can be found in healthy muscles, and an increased frequency of split myofibers is commonly observed in muscular dystrophy and various neurogenic myopathies [111,112]. An increased frequency of split myofibers has also been observed in muscles subjected to mechanical loading. For instance, the most extraordinary example of this was published by Antonio and Gonyea (1994) who reported that the frequency of split myofibers in the quail ALD muscle increased from 0.25% to 5.25% after 28 days of wing-weighting [113]. Tamaki et al. (1996) also found that the frequency of split myofibers in the rat plantaris muscle increased from 0.6% to 1.8% after 6 weeks of synergist ablation [114]. An increase in the occurrence of split myofibers (1.4% of all myofibers) has also been observed in powerlifters that used anabolic steroids [115]. Based on these reports, it would appear mechanical loading can result in an increased prevalence of split myofibers. However, it is important to point out that many of the studies that are frequently cited as providing support for this concept never actually quantified the number of split myofibers [116,117,118]. Moreover, there are multiple examples in which the number of split myofibers was quantified, and it was concluded that mechanical loading did not alter the frequency of their appearance [119,120,121,122]. Even the study by Antonio and Gonyea (1994) found that 16 days of wing-weighting resulted in an 88% increase in the mass of the ALD muscle, yet the frequency of split myofibers at this time point was still only 0.28% [113]. One potential explanation for this observation is that splitting along the entire length of the myofiber rapidly runs to completion, and thus, only a small fraction of the myofibers that split are effectively detected. However, if this were the case, then the total number of myofibers per muscle should increase. To test this, Antonio and Gonyea (1994) directly counted all of the fibers in the ALD muscles and found that the total number did not change after 16 days of wing-weighting [113]. Thus, it is our conviction that although mechanical loading may be capable of inducing myofiber splitting, the frequency of this event is low and thus does not typically make a major contribution to the overall growth process.

#### 4.2.3. Hyperplasia

Hyperplasia refers to the generation of new myofibers, and as illustrated in Figure 4B, hyperplasia could lead to the radial growth of muscle fascicles. Indeed, numerous studies have shown that the number of myofibers per muscle rapidly increases during the early stages of developmental growth [123,124,125]. Although it is well accepted that hyperplasia occurs during developmental growth, whether hyperplasia can occur in adult skeletal muscles remains a subject of debate.

Part of the debate over whether hyperplasia occurs in adult skeletal muscle results from the types of measurements that have been used to address this question [126,127]. Specifically, two primary methods have been employed: (1) counting the number of myofibers per cross-section of the muscle, and (2) digestion of the muscle’s connective tissue followed by a direct count of all myofibers present in the muscle. The direct counting method is ideal, but this approach requires the manual dissociation and counting of thousands of myofibers. Accordingly, most studies that describe measurements of hyperplasia are based on counts of the myofibers per cross-section. With this point in mind, it is imperative to recognize that the number of myofibers that appear in a cross-section can be highly influenced by changes in the architectural properties of the muscle (e.g., fiber length and/or pennation angle) [75,76,91,127]. Such effects have been thoroughly described by Maxwell et al. (1974), and can be appreciated by considering the illustrations presented in Figure 3 and Figure 4 [75]. For instance, Figure 3B shows how an 11% increase in fascicle length would lead to a 30% in the number of fascicles per cross-section, and the same principles would hold at the level of the myofibers. Furthermore, as illustrated in Figure 4C, longitudinal growth of myofibers with intrafascicular terminations could also lead to an increase in the number of myofibers per cross-section. Hence, extreme caution needs to be exercised when interpreting the results from studies that rely on myofiber per cross-section counts to make conclusions about hyperplasia.

Unfortunately, the majority of studies that have examined whether mechanical loading induces hyperplasia have relied on counts of the myofibers per cross-section [22,35,109,116,117,118,128,129,130]; however, there are a handful of studies that have reported direct myofiber counts. For instance, investigators with ties to Dr Gonyea reported that 7–30 days of wing-weighting resulted in a 60–294% increase in muscle mass and a 30–50% increase in the number of myofibers per muscle [72,93,113,131]. In stark contrast, investigators with links to Dr Gollnick reported that 6–65 days of wing-weighting resulted in a 22–225% increase in muscle mass but no change in the number of myofibers per muscle [120]. The Gollnick group also reported that other extreme forms of mechanical loading such as synergist ablation does not alter the number of myofibers per muscle [119,121]. The conflicting conclusions from these groups has existed for over 25 years, and surprisingly, the controversy has still not been resolved [126,127,132,133]. Thus, in our opinion, the notion that extreme forms of mechanical loading can induce hyperplasia remains controversial.

Due to a minimal number of studies, a similar controversy exists with regards to whether more physiologically relevant models of mechanical loading can induce hyperplasia. Indeed, we are only aware of two studies that have directly addressed whether a resistance exercise-like stimulus can alter the number of myofibers per muscle. The first of these studies was performed by Gonyea et al. (1986) whom painstakingly counted the number of myofibers in the flexor carpi radialis of cats that had been subjected to 60–129 weeks of weight training, and the results indicated that the training stimulus led to a 9% increase in the number of myofibers per muscle (39,759 vs. 36,550 myofibers per muscle) [122]. Likewise, Tamaki et al. (1992) subjected rats to weight-lifting exercise and found that the number of myofibers in the plantaris muscle increased by 14% after 12 weeks of training; however, in this case, the absolute mass of the plantaris was not significantly altered by the training, and thus, the basis for the increase in fiber number is difficult to interpret [134]. Mixed and cautious interpretations can also be drawn from studies that have used myofiber per cross-section counts as a readout for hyperplasia, with some studies showing an increase in the number of myofibers per section [110,116,128], while other have reported no change [35,100,110]. Accordingly, a firm conclusion with regards to whether physiologically relevant forms of mechanical loading can induce hyperplasia remains elusive.

## 5. Mechanical Load-Induced Growth of Skeletal Muscle at the Ultrastructural Level

### 5.1. Longitudinal Growth of Myofibers

As summarized in the previous section, a compelling body of literature has shown that extreme models of mechanical loading can promote the longitudinal growth of myofibers. Furthermore, several lines of evidence suggest that longitudinal growth of myofibers can also be induced by physiologically relevant forms of mechanical loading. Since myofibers are composed of an in-series connection of sarcomeres, it follows that an increase in myofiber length would be mediated by an increase in the length of the sarcomeres and/or the serial addition of new sarcomeres. When considering these options, it is essential to bear in mind that the optimal length of sarcomeres (≈2.5 μm) is highly conserved, and most muscles operate within a narrow range of the sarcomeres optimal length (94 ± 13%) [43]. Hence, it can be inferred that a mechanical load-induced increase in myofiber length would most likely be driven by the serial addition of new sarcomeres, as this would allow for the optimal length of the sarcomeres to be maintained in the elongated myofiber.

In support of the above rationale, Williams and Goldspink (1971) demonstrated that the increase in myofiber length that occurs during development is highly correlated with the serial addition of new sarcomeres [51], and a similar relationship is observed during the myofiber lengthening that occurs in response to increased mechanical loading. For instance, Williams and Goldpink (1973) demonstrated that the number of sarcomeres along the length of mouse soleus myofibers increases by 23% after ≈ 7 days of tenotomy (a milder form of the synergist ablation model) [135]. Likewise, Aoki et al. (2009) have shown that the number of sarcomeres along the length of rat soleus myofibers increases by 27% after just 4 days of chronic stretch [54]. Collectively, these, and many other studies [51,54,95,96,97,135,136,137,138,139], have not only indicated that mechanical loading could lead to the serial addition of new sarcomeres but also suggest that this type of growth can occur in a very rapid manner.

If mechanical loading leads to the serial addition of new sarcomeres, then it raises the question of where along the length of the myofibers the new sarcomeres are added. According to Goldspink (1983) “The point or points at which the sarcomeres are added has been rather uncertain until recently. With radioactively labeled amino acids and radioactively labeled adenosine the site of longitudinal growth was shown to be at the ends of the myofibrils” [140]. Although this is a fundamentally important conclusion, its validity remains highly contestable.

The first study that Goldspink cited as providing support for his conclusion was published by Griffin et al. (1971) and used ^3^H-adenosine as a means for labeling where newly synthesized actin was deposited during the postnatal growth of myofibers [141]. Specifically, young mice were injected ^3^H-adenosine, and then single myofibers were imaged with autoradiography. Based on the results, Griffin et al. concluded that the ^3^H-adenosine was primarily deposited at the ends of the myofibers. Importantly, however, this conclusion was not supported by quantitative data, and the images included in the manuscript were far from persuasive [141].

The second study that Goldspink cited as support for his conclusion used ^3^H-adenosine in an effort to identify where new sarcomeres were added in adult soleus muscles that were recovering from being immobilized in a shortened position [135]. The study began with a clear demonstration that serial sarcomere addition occurred during the recovery period. After establishing this point, the muscles were cut into 5 separate regions along the longitudinal axis and then analyzed for ^3^H-adenosine. As shown in Figure 5, the outcomes revealed that the amount of ^3^H-adenosine in the two most distal regions of the muscle was significantly elevated in muscles that were undergoing recovery. Importantly, however, whether the enhanced ^3^H-adenosine deposition was due to formation of new sarcomeres at the ends of the myofibrils was not directly tested. Indeed, it could be argued that the results from this study simply reflect the type of regional differences in the mechanical load-induced growth that we described in Section 3.1.

In contrast to the notion that new sarcomeres are added at the ends of the myofibrils, others have provided evidence which suggests that new sarcomeres can be inserted throughout the length of the myofibrils [118,143,144,145,146,147,148,149,150]. For instance, when studying the developmental growth of single myofibers that possess two separate motor endplates, Bennett et al. (1985) discovered that the distance between the motor endplates increased in a manner that was directly proportional to the increase in myofiber length [146]. Similar evidence was obtained by Mackay and Harrop (1969) whom inserted wire markers at various points along the length of the sternomastoid and anterior gracilis muscles of 4 week old rats and then tracked their position with x-ray images during the subsequent 8 weeks of developmental growth [145]. In this case, a proportionate increase in the distance between wires occurred as the muscles grew in length, and this led the authors to conclude that the myofibers “must be adding new material at points all along their length as they grow”. Indeed, Jahromi and Charlton (1979) obtained support for this concept when they found evidence of a longitudinal growth process that appears to involve the transverse splitting of sarcomeres that are embedded within the midst of the myofibrils (Figure 6A) [143].

As described by Jahromi and Charlton (1979), the transverse splitting of sarcomeres appears to occur through an ordered sequence of events which include: (1) splitting of the thick filaments at the H-zone, (2) elongation of the two halves of the thick filaments along with the formation of new thin filaments in the previous H-zone, and (3) formation of a new Z-disc in the center of the newly formed thin filaments [143]. Although this process was originally described in crab skeletal muscles, there is evidence to suggest that the same process takes place in vertebrates. For instance, as shown in Figure 6B, Vaughan and Goldspink (1979) observed a similar phenomenon in soleus muscles of mice that had been subjected synergist ablation; however, in this instance, it was thought that the splitting was reflective of damage to the sarcomeres [118]. In fact, focal disruptions of the sarcomere, such as lesions, Z-disc streaming, and Z-disc smearing have long been viewed as markers of damage [152,153,154,155,156]. However, as detailed in a series of publications by Yu et al., these regions might simply be areas of remodeling that result in new sarcomere formation [147,148,149]. For instance, when examining soleus muscles from humans that had engaged in a bout of intense eccentric contractions, Yu et al. detected a 5-fold increase in the appearance of regions with “supernumerary sarcomeres” (Figure 6C,D) [147]. Such regions are remarkably similar to the “sphenode” regions that were described by Heidenhain over 100 years ago, which are characterized by the presence of additional sarcomeres that are out of register with the surrounding sarcomeres [151]. Interestingly, these regions appear to include areas that resemble H-zone transverse sarcomere splitting, as well as another potential type of transverse sarcomere splitting that occurs at the Z-disc (Figure 6F) [150]. Thus, when considering the studies that have been highlighted in this section, it is fair to conclude that mechanical loading can lead to the longitudinal growth of myofibers and this process is primarily driven by the serial addition of new sarcomeres. However, exactly how and where new sarcomeres get added along the length of the myofibrils remains to be resolved.

### 5.2. Radial Growth of Myofibers

In Section 4.2.1 we reviewed the evidence which indicates that the radial growth of myofibers (i.e., myofiber hypertrophy) is one of, if not the, primary contributor to the growth that occurs in response to increased mechanical loading. We will now examine what is known about the ultrastructural adaptations that drive this process. However, before going deeper into this topic, it is important to consider the concept of specific tension, which is defined as the maximal isometric force produced per CSA. At the myofiber level, the underlying premise for this concept is that the maximal isometric force is directly dependent on the number of the force-generating elements that act in parallel with the line of force production, and that the number of these elements is directly dependent on the CSA of the myofiber [157,158]. This thesis becomes particularly important when formulating hypotheses about the mechanisms that potentially contribute to the radial growth of the myofibers. For instance, if the CSA of a myofiber increases and specific tension remains constant, then it can be inferred that the radial growth was due to a proportionate addition of both force-generating elements (e.g., myofilaments/sarcomeres/myofibrils) and non-force-generating elements (e.g., mitochondria, sarcoplasmic reticulum, intracellular fluid, connective tissue, etc.). Alternatively, if the CSA of a myofiber increases and specific tension decreases, then it can be inferred that the radial growth was due to a disproportionately greater increase in the amount of non-force-generating elements. Thus, through measurements of specific tension, one can obtain fundamental insight into the mechanisms that drive the radial growth of the myofibers.

As summarized in a recent meta-analysis by Dankel et al. (2019), at least 15 different studies have assessed whether the specific tension of individual myofibers is impacted by resistance exercise. Importantly, the overwhelming majority of these studies have concluded that specific tension is either not significantly altered, or slightly increases in hypertrophied myofibers [159,160,161,162,163,164,165,166,167]. Similar observations have also been made in myofibers that were isolated from muscles that have adapted to extreme forms of mechanical loading, such as synergist ablation [168]. Thus, it would appear that the radial growth of myofibers is driven by a proportional increase in the force-generating and non-force-generating elements. However, despite this evidence, some have argued that a disproportionate increase in the non-force-generating elements can make a substantive contribution to radial growth. This type of radial growth has generically been referred to as sarcoplasmic hypertrophy, and in the following sections we will address in greater detail whether radial myofiber growth is driven by sarcoplasmic hypertrophy and/or the expansion of the force-generating elements that act in parallel with the line of force production.

#### 5.2.1. Sarcoplasmic Hypertrophy

Anecdotal observations suggest that although bodybuilders have bigger muscles than powerlifters, they are not as strong. Such observations have led many to contend that myofiber hypertrophy in bodybuilders is due to a disproportionately larger increase in non-force-generating elements (i.e., sarcoplasmic hypertrophy). It has also been hypothesized that these non-force-generating elements could include osmotically active metabolites (e.g., creatine and glycogen) that would draw water into the myofiber, and/or organelles such as the sarcoplasmic reticulum and mitochondria [101,169,170]. However, the relevance of these hypotheses is dependent on whether sarcoplasmic hypertrophy makes a substantive contribution to the mechanical load-induced growth of myofibers. Thus, in this section, we will critically evaluate the evidence that surrounds this concept.

Several studies have been commonly cited as providing support for the existence of sarcoplasmic hypertrophy [32,36,163,171,172,173,174]. For instance, D’Antona et al. (2006) measured specific tension in single myofibers from recreationally active subjects, and from subjects that had engaged in bodybuilding for at least 2 years. With regards to providing support for sarcoplasmic hypertrophy, the often-cited outcome is that specific tension was lower in the Type I fibers of bodybuilders [163]. However, it is important to point out that the same study also observed an increase in specific tension of the Type IIA and IIX myofibers from the same bodybuilders [163]. The work of Meijer et al. (2015) is another frequently cited study that measured specific tension in single myofibers. In this case, specific tension was measured in myofibers from control subjects, bodybuilders, and powerlifters. Importantly, it was concluded that specific tension was lower in the myofibers obtained from bodybuilders [172]. At first glance it would appear that this study provides clear support for the notion that bodybuilders experience sarcoplasmic hypertrophy; however, 9 of the 12 bodybuilders in the study admitted to recent use of anabolic steroids [172]. This is noteworthy because the use of anabolic steroids has been associated with alterations in protein composition and the morphological properties of myofibers [175,176]. Indeed, MacDougall et al. (1982) reported a 9.8% decrease in the proportion of the myofiber CSA that is occupied by the myofibrils in elite bodybuilders and powerlifters (6 of 7 of whom admitted to the use of anabolic steroids), whereas only a 1.6% difference was observed after 6 months of resistance exercise in subjects that denied the use of anabolic steroids [32]. In addition to the aforementioned concerns, it also bears mentioning that the studies by D’Antona et al. and Meijer et al. were both cross-sectional in nature. This is important because it is well known that cross-sectional studies cannot be used to infer cause and effect relationships [177,178,179,180]. Thus, caution needs to be used when considering whether the outcomes of D’Antona et al. and Meijer et al. provide support for the presence of sarcoplasmic hypertrophy.

Other studies that have been cited as providing support for the existence of sarcoplasmic hypertrophy include the work of Penman (1969) who subjected participants to 8 weeks of an exercise intervention that included either progressive resistance exercise, isometric contractions, or stair running [171]. The frequently cited outcome from this study is that exercise led to a decrease in the “myosin concentration” (defined as number of myofibrils in a 5 µm^2^ area) [171]. However, this study only included 2 subjects per group, there was a substantial amount of variance in the data, and no statistical analyses were performed.

Another commonly cited study involves the work of Toth et al. (2012) whom subjected older subjects (≈73 years of age) to 18 weeks of resistance exercise and observed a significant decrease in the proportion of the myofiber CSA that was occupied by the myofibrils [36]. Importantly, however, the resistance exercise program employed in this study did not lead to a significant increase in myofiber CSA. Thus, if anything, the observed decrease in the proportion of the CSA that was occupied by the myofibrils would suggest that the resistance exercise program led to the selective loss of the myofibrils rather than a disproportionately large increase in non-force-generating elements (i.e., sarcoplasmic hypertrophy).

More recently, Haun et al. (2019) concluded that the myofiber hypertrophy that occurs after 6 weeks of high-volume resistance training can be largely attributed to sarcoplasmic hypertrophy [173]. Specifically, the key piece of evidence in this study was the observed trend for a decrease in the concentrations of myosin and actin after the 6 weeks of training (P = 0.052 and P = 0.055, respectively) [173]. Although these results are interesting, it should be noted that 15 subjects were analyzed in this study, and they only represented a subset of the 31 subjects that participated in the original training intervention [180]. More importantly, the 15 subjects that were examined only included the subjects who showed an “increase” in myofiber CSA (responders by the authors’ definition) [173]. This is important because when all 31 subjects from the original training intervention were considered, it was determined that the 6 weeks of training did not induce myofiber hypertrophy [180]. Accordingly, the results of Haun et al. (2019) cannot be viewed as being representative of the whole population and, are therefore, difficult to interpret within the context of whether sarcoplasmic hypertrophy normally makes a substantive contribution to the mechanical load-induced growth of myofibers.

In summary, we remind the reader that as summarized by Dankel et al. (2019), a large number of longitudinal studies have shown that specific tension is preserved in myofibers that have experienced radial growth as a result of increased mechanical loading [159,160,161,162,163,164,165,166,167,168]. This consistent body of evidence strongly suggests that the radial growth of myofibers is not driven by sarcoplasmic hypertrophy, but rather is due to a proportionate increase in the force-generating and non-force-generating elements that act in parallel with the line of force production.

#### 5.2.2. Expansion of the Force-Generating Elements

In myofibers from vertebrates, the force-generating myofilaments are contained within the sarcomere and organized into a hexagonal array of thick and thin myofilaments [181]. The overall geometry and spacing between the myofilaments is highly conserved, and thus, any changes in the number of force-generating myofilaments that are aligned in parallel would likely be matched by a proportionate alteration in the CSA that is occupied by the sarcomeres/myofibrils [182,183]. Given that specific tension is preserved in myofibers that have experienced radial growth as a result of increased mechanical loading, and that specific tension is dependent on the number of in parallel force-generating elements, it would follow that the radial growth is mediated by a propionate increase in the CSA that is occupied by the sarcomeres/myofibrils. Indeed, a handful of studies have directly tested this thesis, and all of them reported that induction of myofiber hypertrophy was associated with minimal changes (≤4%) in the relative proportion of the CSA that was occupied by the myofibrils [32,33,34,35,184]. For instance, MacDougall et al. (1982) reported that 6 months of resistance exercise in humans led to a 22–25% increase in the CSA of myofibers along with almost no change in the proportion of the CSA that was occupied by the myofibrils (84.2% vs. 82.6% in the pre- and post-trained states, respectively) [32]. Put differently, the data from MacDougall et al. indicated that the total area occupied by the myofibrils increased by ≈23%, but whether this was due to radial growth of the pre-existing myofibrils (myofibril hypertrophy) and/or an increase the number of myofibrils (myofibril hyperplasia) was not determined (Figure 7A) [32]. In fact, we are not aware of any studies that have systematically addressed whether mechanical load-induced myofiber hypertrophy is mediated by myofibril hypertrophy and/or myofibril hyperplasia. In our opinion, it is easy to envision how the induction of myofibril hypertrophy and/or myofibril hyperplasia could serve as the foundational events by which mechanical loading drives the radial growth of myofibers, thus the lack of knowledge on this topic is quite surprising.

Even though the concepts of myofibril hypertrophy and myofibril hyperplasia have not been thoroughly examined within the confines of mechanical load-induced skeletal muscle growth, there is still much that can be learned from related fields of study (e.g., developmental growth of skeletal muscle, mechanical load-induced growth of the heart, etc.). For instance, seminal work by Goldspink (1970) used mice of various ages to establish that a positive linear relationship exists between myofibril diameter and myofiber CSA, and a similar relationship was also found to exist between myofibril number and myofiber CSA (Figure 7B,C) [185]. Collectively, the results of this study provided some of the first evidence that both myofibril hypertrophy and myofibril hyperplasia could contribute to the radial growth of myofibers. Moreover, these observations provided the basis for Dr Goldspink’s intriguing model of radial growth which involves a process he called myofibril splitting [141,185,186,187]. Specifically, Dr Goldspink proposed that the increase in myofibril number that occurs during the radial growth of myofibers could be explained by the longitudinal splitting of pre-existing myofibrils. In support of his hypothesis, he published numerous longitudinal images of single myofibrils that appeared to split into two smaller daughter myofibrils (Figure 8A) [185,186,187]. Moreover, he demonstrated that the splits usually occurred in the middle of Z-disc, and were typically found in myofibrils that are twice as large as myofibrils that did not contain splits [185].

In addition to his observations on longitudinal splitting, Dr Goldspink also noted that the thin myofilaments in sarcomeres do not run directly perpendicular to the Z-disc, but instead are offset at a slightly oblique angle (≈6–10°) [186]. This was an important observation because it suggested that the thin myofilaments could exert outward radial forces on the Z-disc when the sarcomeres contract. Indeed, this became a key part of his myofibril splitting model in which it was proposed that myofibrils initially undergo hypertrophy and, as their diameter increases, the outward radial forces that they exert on the Z-disc also increases. The outward radial forces place a strain on the center of the Z-disc, and when these forces reach a critical threshold, it causes the Z-disc to break (Figure 8B). The break begins at the center of the Z-disc and forms a split which then propagates through the remainder of the myofibril and ultimately forms two smaller daughter myofibrils.

Dr Goldspink’s model of myofibril splitting was developed over 40 years ago, and it has frequently served as the textbook explanation of how myofibril number could increase during the radial growth of myofibers [188,189,190,191]. However, despite being widely accepted, the validity of the model has not been rigorously tested. For instance, we are not aware of any direct evidence that a single myofibril can split into daughter myofibrils. Furthermore, we are not aware of any studies that have established whether the outward radial forces generated by the obliquely aligned myofilaments would be physically capable of “breaking” the Z-disc. In addition to limited evidence, there are also parts of myofibril splitting model that seem to be incomplete. For instance, as shown in Figure 8B, it has been shown that the diameter of the myofibrils is directly related to the size of the myofibers, and from our point of view, Dr Goldspink’s model is not capable of explaining this relationship [185]. Nevertheless, the general concepts of the myofibril splitting model are well reasoned and, as such, it will serve as framework for remainder of our discussions on myofibril hypertrophy and myofibril hyperplasia.

#### 5.2.3. Myofibril Hypertrophy

If we assume that the basic concepts of the myofibril splitting model are correct, and that they can be applied to the radial growth of myofibers that occurs in response to increased mechanical loading, then the first part of the overall growth process would involve myofibril hypertrophy. This initial hypertrophic response would continue until the myofibrils reached the critical size that induces splitting. The splitting would result in the formation of daughter myofibrils that would then undergo hypertrophy until they split, and the cycle would repeat until the radial growth of the myofiber commenced. We will now examine the limited body of literature that surrounds this thesis.

To the best of our knowledge, only one study has addressed whether mechanical load-induced myofiber hypertrophy is associated with myofibril hypertrophy [184]. This study was performed by Ashmore and Summers (1981) and was focused on defining the changes that occur in the patigialis muscle of young chickens after 1–7 days of wing-weighting [184]. Importantly, the same group had previously demonstrated their model of wing-weighting leads to an ≈55% in myofiber CSA after 7 days [136], and not surprisingly, their 1981 publication revealed that the increase in myofiber CSA was matched by a proportionate increase in the CSA that was occupied by the myofibrils [185]. In this study, they also found that the average CSA of the individual myofibrils increased by 36% after 7 days, and this was associated with a 2.6-fold increase in the proportion of myofibrils that presented with signs of splitting [136]. When taken together these results are very noteworthy because they provide critical support for the notion that mechanical loading can induce myofibril hypertrophy, and that this effect is associated with an increase in myofibril splitting.

The results of Ashmore and Summers (1981) provided support for the notion that mechanical loading can induce myofibril hypertrophy, and therefore raise questions about the processes that drive this response [184]. When considering these processes it is important to remember that the force-generating myofilaments within the myofibrils are organized into a hexagonal array and the spacing between the myofilaments is highly conserved [181,182,183]. Thus, it can be predicted that an increase in the CSA of the myofibril would be met by a proportionate increase in the number of force-generating myofilaments per CSA. If this is correct, then one is left with the question of where the new myofilaments get deposited.

As illustrated in Figure 9, some possible locations of new myofilament deposition include but are not limited to: (A) the periphery of the pre-existing myofibril, (B) the center of the pre-existing myofibril, or (C) throughout the pre-existing myofibril. All these options seem plausible, but options B and C would likely require extensive remodeling of the pre-existing myofilament lattice, whereas option A presumably would not. Thus, from a resources/energetic standpoint, the deposition of new myofilaments at the periphery of the pre-existing myofibril would appear to be the most cost-effective and least disruptive option.

The work of Morkin (1970) is often cited as providing support for the notion that new myofilaments are added to the periphery of myofibrils [192]. Specifically, in this study, rat diaphragm muscles were incubated with ^3^H-leucine to label newly synthesized proteins, and then electron microscope autoradiography was used to identify the location of the newly synthesized proteins [192]. As shown in Figure 10A, the location of the newly synthesized proteins was indicated by the presence of relatively large (≈300 nm) electron dense grains. The quantitative results from this study are shown in Figure 10B with the bars indicating how frequently the center of the grains appeared at various distances from the periphery of the myofibril, and the green highlighted curve illustrating the theoretical distribution of the grains that would be expected if the myofibrils were labeled exclusively at the periphery. At first glance, the close match between the theoretical and observed values appears to provide compelling support for the conclusion that new myofilaments are added to the periphery of the myofibrils [192]. However, this evidence becomes less persuasive when one considers that ribosomes are typically localized in the intermyofibrillar space and many of these ribosomes appear in polysomal configurations which is indicative of active protein synthesis (Figure 10C) [193,194]. This point leads us to question how well the data from Morkin (1970) would fit with a different hypothesis. In this case, the hypothesis was that the ribosomes in the intermyofibrillar space are actively engaged in the synthesis of new proteins. In Figure 10D,E, we have illustrated how well the data from Morkin (1970) fit with the theoretical distribution of the grains that would be expected if the myofibrils were labeled exclusively at the periphery, and compared that with the theoretical distribution of the grains that would be expected if newly synthesized proteins were located exclusively within the intermyofibrillar space. The key point from this illustration is that the data appears to be consistent with both theoretical distributions, and this is because the resolution provided by autoradiography simply does not allow for a clear distinction between the two possibilities.

The limitations of the resolution that can be obtained with electron microscope autoradiography have been thoroughly described by Caro (1962) and Salpeter et al. (1969) [195,198]. Importantly, both of these studies demonstrate that under typical conditions, 50% of the grains will develop within ≈130 nm of the source and 95% of the grains will develop within ≈300 nm [195,198] (Figure 10F). This level of resolution would be outstanding if the goal was to identify the location of newly synthesized proteins within a myofiber (typical diameter of 25,000 nm), but it is far from ideal when the goal is to identify the location of newly synthesized proteins within a myofibril (typical diameter 850 nm). To effectively accomplish this goal, technologies that offer a much higher level of resolution are needed, and fortunately, such technologies are now available. For instance, it is now possible to identify the location of newly synthesized proteins with immunological and click-chemistry-based technologies [199,200]. This is noteworthy because, as illustrated in Figure 10F, a typical immunoelectron microscopy-based approach will result in 100% of the signal appearing within 20 nm of the source, and the use of more advanced approaches (e.g., 1 nm gold conjugated Fab antibody fragments, or click-chemistry-based linkers) can allow for a resolution of less than 7 nm [197,201,202,203]. Thus, although we still do not know whether mechanical load-induced hypertrophy of myofibers is driven by myofibril hypertrophy, or where new myofilaments get added during the process of myofibril hypertrophy, the technologies that are need to answer these fundamental questions are now within our reach.

#### 5.2.4. Myofibril Hyperplasia

As mentioned in the previous section, the study by Ashmore and Summers (1981) provided support for the notion that the mechanical load-induced radial growth of myofibers is associated with myofibril hypertrophy, but unfortunately, the study did not address the concept of myofibril hyperplasia [184]. In fact, we are not aware of any studies that have directly addressed this concept, and the only study we could find that even came remotely close was performed by Holmes and Rasch (1958) [204]. Specifically, this study involved 7 weeks of training rats with progressively more intense running and concluded that the number of myofibrils per myofiber in the sartorius muscle was not significantly altered by the training regime [204]. However, it was also determined that the training regime did not lead to a significant increase in mass of the sartorius muscle and, thus, it is difficult to extrapolate any meaningful insights from the data.

Although we are not aware of any studies in skeletal muscle that have directly addressed whether mechanical load-induced myofiber hypertrophy is associated with myofibril hyperplasia, there are a few studies that have addressed this topic in the heart. For instance, Toffolo and Ianuzzo (1994) used aortic constriction to subject rat hearts to mechanical overload and found that after 30 days, the cardiomyocyte area had increased by ≈50% and this was associated with an ≈70% increase in the number of myofibrils per cardiomyocyte [205]. An increase in the number of myofibrils per cardiomyocyte has also been observed in hypertrophied human hearts that were examined postmortem [206]. Furthermore, Anversa et al. (1980) examined heart papillary muscles after 8 days of mechanical overload and observed a 55% increase in the CSA of the cardiomyocytes that was occupied by the myofibrils, but no change in the CSA of the individual myofibrils, thus implying an increase in myofibril number [207]. Taken together, these studies consistently suggest that an increase in mechanical loading can lead to an increase in myofibril number in the heart, but whether the same effect occurs in skeletal muscles remains to be determined.

#### 5.2.5. The Radial Growth of Myofibers—Closing Remarks

As we have discussed, a substantial body of evidence indicates that the mechanical load-induced radial growth of myofibers is mediated by a proportional increase in the force-generating and non-force-generating elements. The force-generating elements are contained within the myofibrils, and the myofibrils account for ≈80% of the myofiber CSA. Thus, it can be argued that the bulk of the radial growth is driven by an expansion of the myofibrils. However, whether this expansion is due to hypertrophy of the individual myofibrils and/or myofibril hyperplasia remains to be established. Based on our collective view of the literature, we propose that both processes are involved, and can be explained by a model that we have defined as the “myofibril expansion cycle”. Specifically, as illustrated in Figure 11, the myofibril expansion cycle begins with the deposition of new myofilaments around the periphery of the pre-existing myofibrils, and results in myofibril hypertrophy. Once the myofibrils reach a critical size, they split and subsequently form two smaller daughter myofibrils. The daughter myofibrils are then able to enter another round of the cycle, and the cycle repeats until the radial growth of the myofiber has commenced. Clearly, our model is based on an integration of hypotheses that were proposed more than 40 years ago, and as emphasized throughout this section, the validity of these hypotheses have not been rigorously tested. Fortunately, the technologies that are needed to test these hypotheses are now available. Thus, we hope that this section will help to inspire new investigations into this seemingly forgotten, yet critically important aspect of skeletal muscle biology.

## 6. Take Home Messages

Mechanical loads are one of the most potent regulators of muscle mass and the maintenance of muscle mass plays a critical role in health and quality of life. In Table 1 we have summarized the major structural adaptations that have been implicated in the mechanical load-induced growth of skeletal muscle. Based on our review, we have also considered whether each of these adaptations makes a substantive contribution to the overall growth process, as well as the level of evidence that is available to support that conclusion. The table also lists some of the major gaps in knowledge that we identified during our review of the literature. Importantly, this is not meant to be an exhaustive summary, and exclusion from the table does not indicate that a given adaptation or gap in knowledge is unimportant (e.g., satellite cell fusion, are satellite cells necessary for mechanical load-induced growth, etc.).

As documented in this review, several of the adaptations that we consider as having weak supporting evidence have been engrained in the literature as “textbook” mechanisms (e.g., the longitudinal growth of myofibers is driven by the addition of new sarcomeres at the ends of myofibers, new myofibrils are formed via myofibril splitting, etc.). We hope that after reading this review, the reader appreciates how little we actually know about the structural adaptations that drive skeletal muscle growth, and the number of extremely fundamental gaps in knowledge that remain to be filled.

## Figures and Tables

**Figure 1 cells-09-01658-f001:**
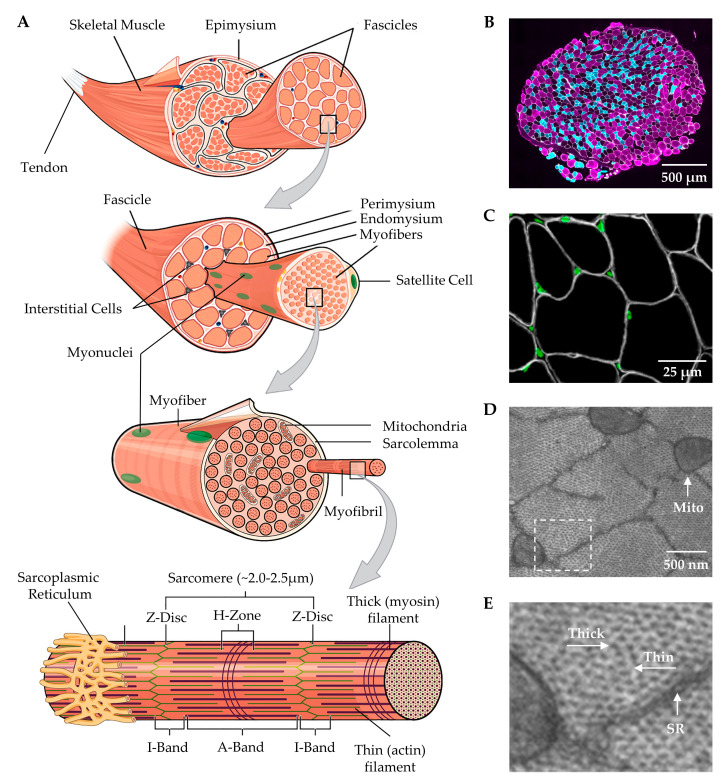
(**A**) Illustration of skeletal muscle structure copied with permission under the Creative Commons Attribution 4.0 International license and adapted for this review, available online: https://openstax.org/books/anatomy-and-physiology/pages/10-2-skeletal-muscle (accessed on 6 January 2020) [28]. (**B**) Cross-section of a mouse plantaris muscle that was subjected to immunohistochemistry for the identification of Type IIA (cyan), and Type IIB (magenta) myofibers as well as laminin to identify the endomysium (white). (**C**) Cross-section of a mouse plantaris muscle that was subjected to immunohistochemistry for the identification of dystrophin to identify the inner boundary of the sarcolemma (white) and nuclei (green). (**D**) Cross-section of a mouse plantaris muscle that was subjected to electron microscopy to highlight the sarcoplasmic reticulum (SR) that surrounds individual myofibrils as well as the mitochondria (Mito) that run between the myofibrils. (**E**) Higher magnification of the boxed region in D reveals the presence of the thick and thin myofilaments.

**Figure 2 cells-09-01658-f002:**
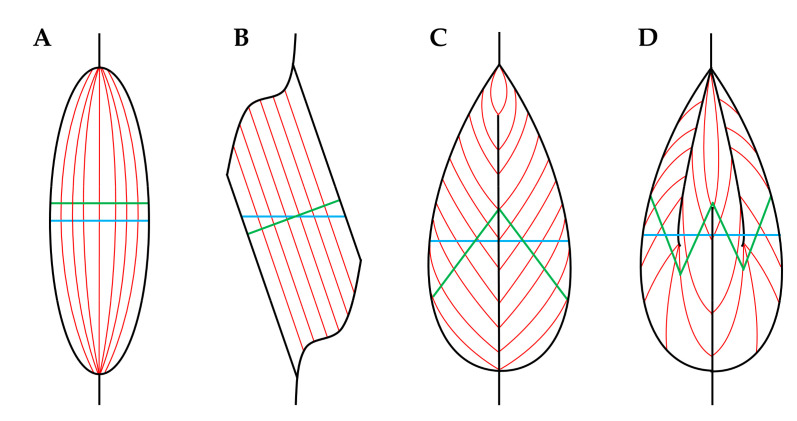
Various pennation angle arrangements of the fascicles/myofibers in skeletal muscle: (**A**) fusiform, (**B**) unipennate, (**C**) bipennante, (**D**) multipennate. Blue lines indicate the plane for the anatomical cross-sectional area (CSA) (i.e., the CSA that runs perpendicular to the longitudinal axis of the muscle), and green lines indicate the plane for physiological CSA (i.e., the CSA that runs perpendicular to the longitudinal axis of the fascicles/myofibers). Adapted under the Creative Commons Attribution-Share Alike 3.0 Unported license from original work by Uwe Gille (Available online: https://creativecommons.org/licenses/by-sa/3.0/deed.en (accessed on 6 January 2020).

**Figure 3 cells-09-01658-f003:**
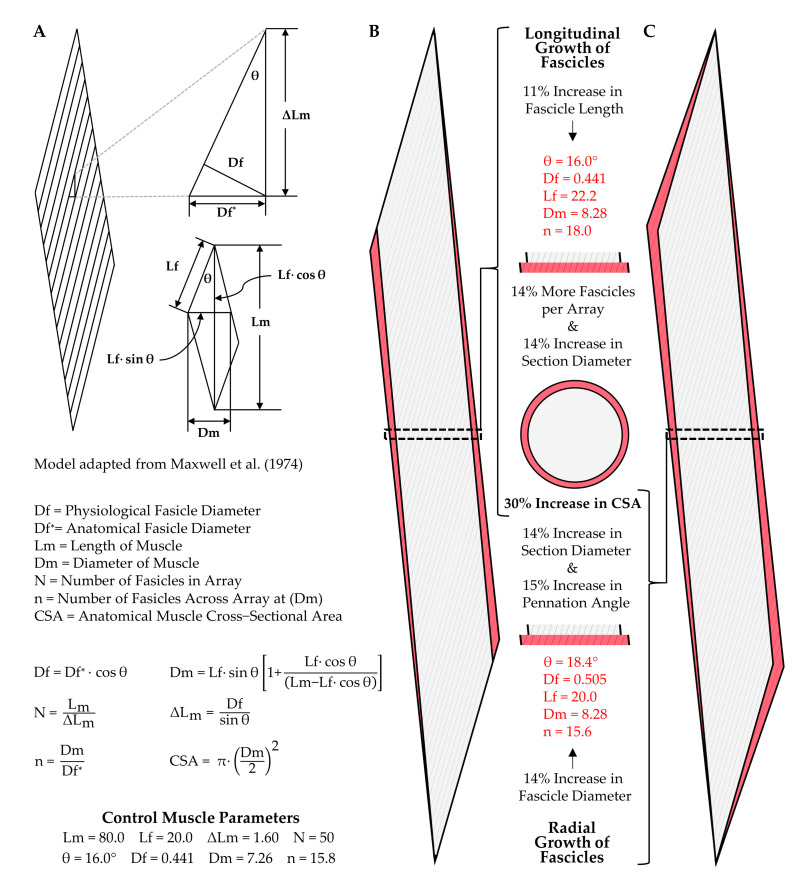
Illustration of how the longitudinal and radial growth of fascicles can lead to changes in muscle cross-sectional area (CSA). (**A**) Key elements of a geometric model that can be used to predict the architectural properties of skeletal muscle [75]. (**B**) Illustration of how an 11% increase in fascicle length would result in 30% increase in CSA, as well as a 30% increase in the number of fascicles per cross-section. (**C**) Illustration of how a 14% in fascicle diameter would lead to a 15% increase in the pennation angle and a 30% increase in the CSA, but essentially no change in the number of fascicles per cross-section.

**Figure 4 cells-09-01658-f004:**
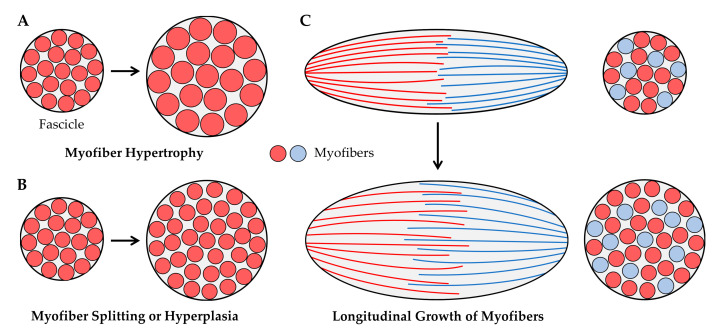
Illustration of how the radial growth of muscles fascicles could result from (**A**) myofiber hypertrophy, (**B**) myofiber splitting or hyperplasia, or (**C**) longitudinal growth of myofibers that exhibit intrafascicular terminations, such as those observed in the long sartorius and gracilis muscles of humans [90].

**Figure 5 cells-09-01658-f005:**
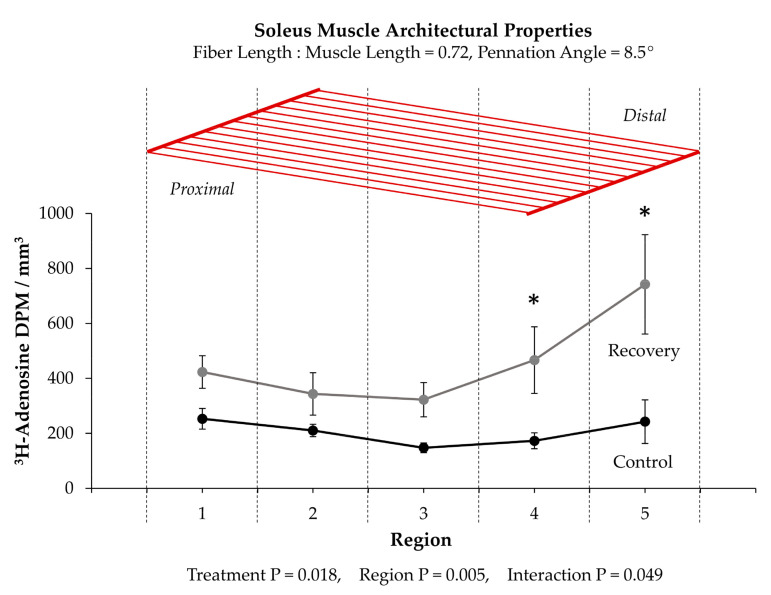
(Top) Schematic illustration of the soleus muscle and its basic architectural properties [142]. (Bottom) Summary of the data provided by Williams and Goldspink (1973) [135]. Values are presented as the means ± SEM and were analyzed with 2-way repeated measures ANOVA. *p*-values for the main effects (i.e., Treatment and Region) and interaction are provided. * Significantly different from the region-matched control condition.

**Figure 6 cells-09-01658-f006:**
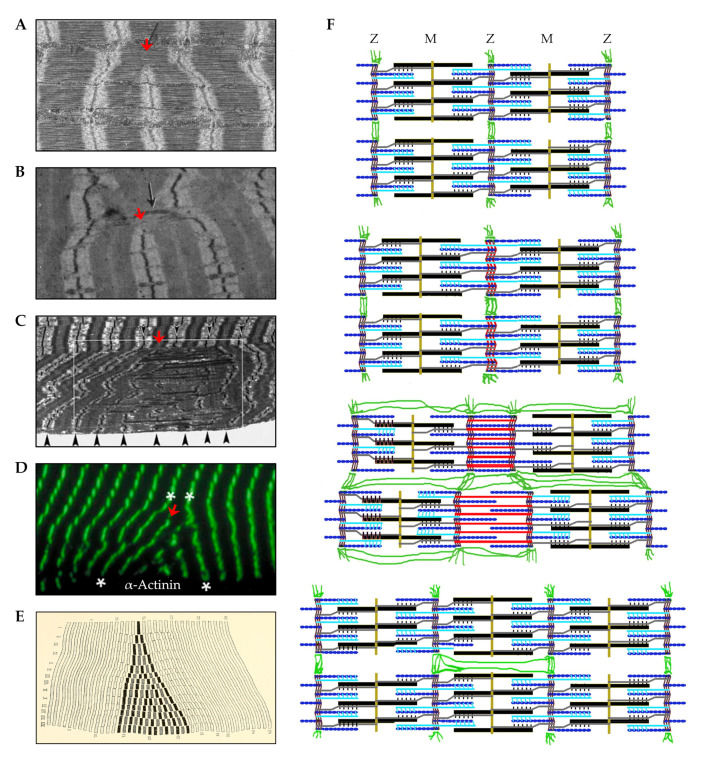
(**A**) Electron micrograph (EM) image that shows a group of myofibrils along with sarcomeres embedded within these myofibrils that possess a transverse split (red arrow) at the H-zone. The image was copied with permission under a Creative Commons License Attribution–Noncommercial–Share Alike 4.0 Unported license, and is available online at https://www.ncbi.nlm.nih.gov/pmc/articles/PMC2110374/ (accessed on 6/20/2020) [143]. (**B**) EM image of myofibrils from a muscle that was subjected to synergist ablation and appears to possess a transverse split at the H-zone (copied with permission from [118]). (**C**,**D**) EM image (**C**) and immunohistochemical image (**D**) of regions with “supernumerary sarcomeres” that are found in human skeletal muscles several days after being subjected to a bout of eccentric contractions (copied with permission from [147,148]). (**E**) Depiction of a “sphenode” region as detailed by Heidenhain (1919) [151]. (**F**) Illustration describing a mechanism for the in-series addition of new sarcomeres via transverse splitting at the Z-disc (copied with permission from [150]).

**Figure 7 cells-09-01658-f007:**
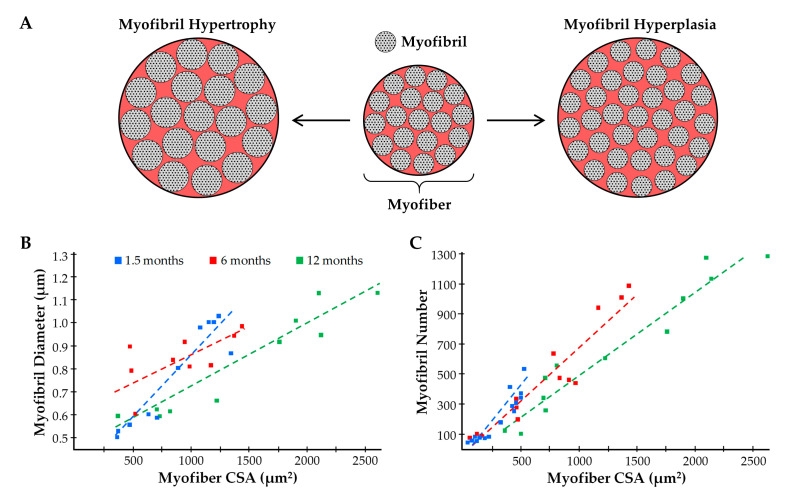
(**A**) Illustration of how an increase in the CSA of the pre-existing myofibrils (myofibril hypertrophy) and an increase in the number of myofibrils (myofibril hyperplasia) can contribute to the radial growth of myofibers. (**B**,**C**) Summary of the data from Goldspink (1970) which highlights the relationship that exists between myofiber CSA and myofibril diameter (**B**), as well as myofiber CSA and myofibril number (**C**), in mice of various ages [185].

**Figure 8 cells-09-01658-f008:**
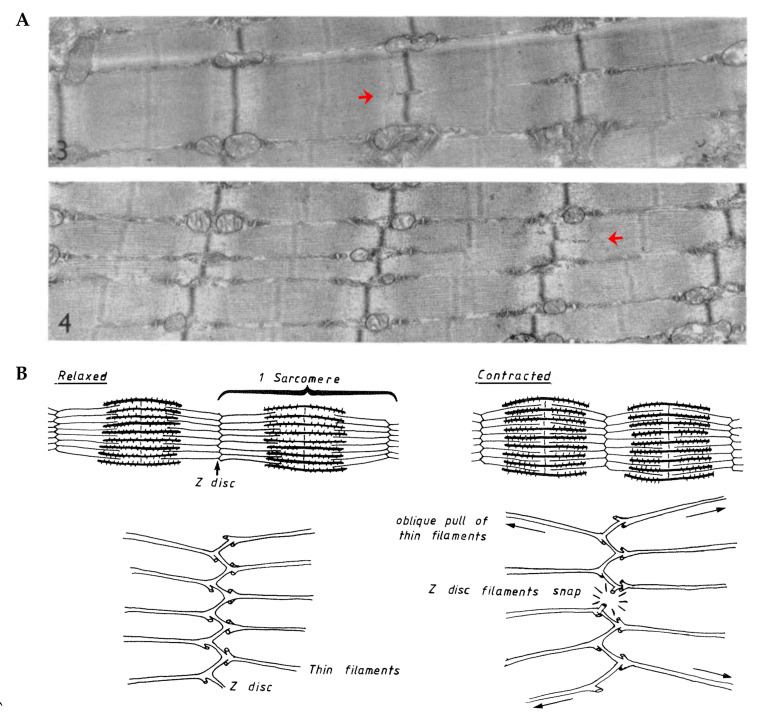
(**A**) Electron micrographs of longitudinal sections from mouse skeletal muscle. Red arrows highlight myofibrils that appear to split into two smaller daughter myofibrils (copied with permission from [186]). (**B**) Illustration from Goldspink (1983) which describes how the oblique angle of the thin myofilaments could exert outward radial forces on the Z-disc when the sarcomeres contract (copied with permission from [140]).

**Figure 9 cells-09-01658-f009:**
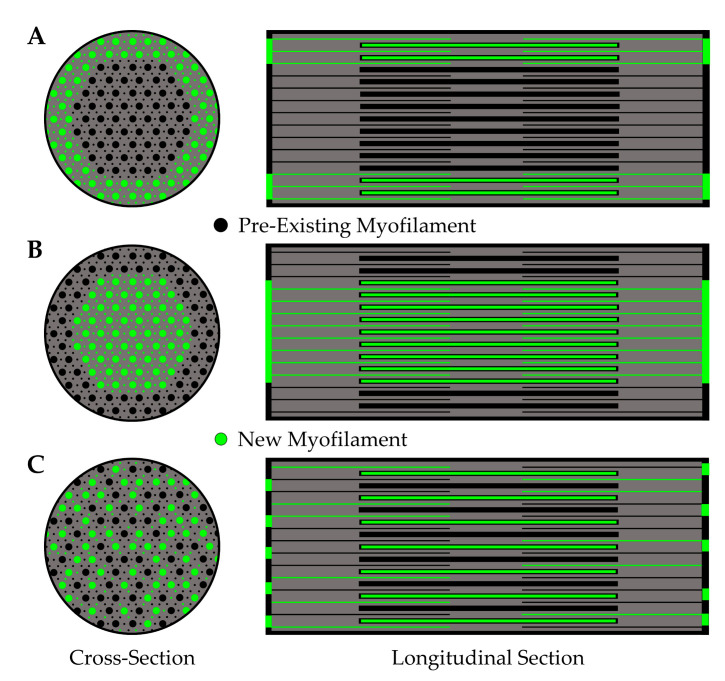
Illustration of where new myofilaments might be added during myofibril hypertrophy. The described possibilities include: (**A**) the periphery of the pre-existing myofibril, (**B**) the center of the pre-existing myofibril, or (**C**) throughout the pre-existing myofibril.

**Figure 10 cells-09-01658-f010:**
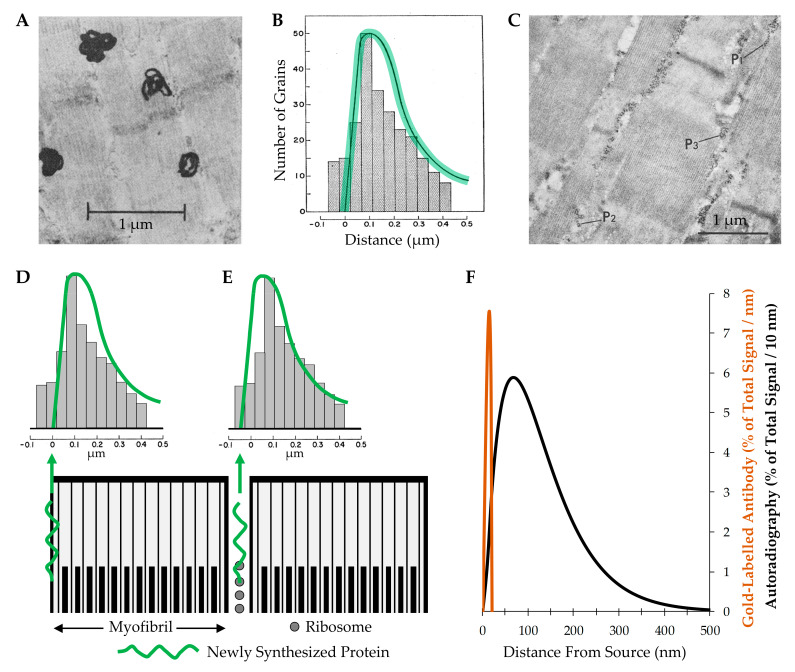
(**A**) Electron microscope autoradiograph from Morkin (1970) which shows the large electron dense grains that were used to identify the location of newly synthesized proteins in the rat diaphragm. (**B**) Bars represent the frequency distribution of the grains relative to the periphery of the myofibril, and the green shaded curve illustrates the theoretical distribution that would be expected if the newly synthesized proteins were located exclusively at the periphery of the myofibril. The images in both A and B were copied with permission from [192]. (**C**) Electron micrograph of the levator ani muscle from an adult rat which reveals the presence of ribosomes in the intermyofibrillar space. Please note that many of the ribosomes appear in different polysomal configurations (P1, P2 and P3) (copied with permission from [193]). (**D**,**E**) Illustration of how well the data from Morkin 1970 fit with the theoretical distribution that would be expected if the newly synthesized proteins were located exclusively at the periphery of the myofibril (**D**) versus being located exclusively within the intermyofibrillar space (**E**). (**F**) A graph illustrating the theoretical radial distribution of the signal obtained with electron microscope autoradiography versus with immunoelectron microscopy that employed a primary antibody (15 nm diameter) conjugated to a 10 nm gold-particle [195,196,197].

**Figure 11 cells-09-01658-f011:**
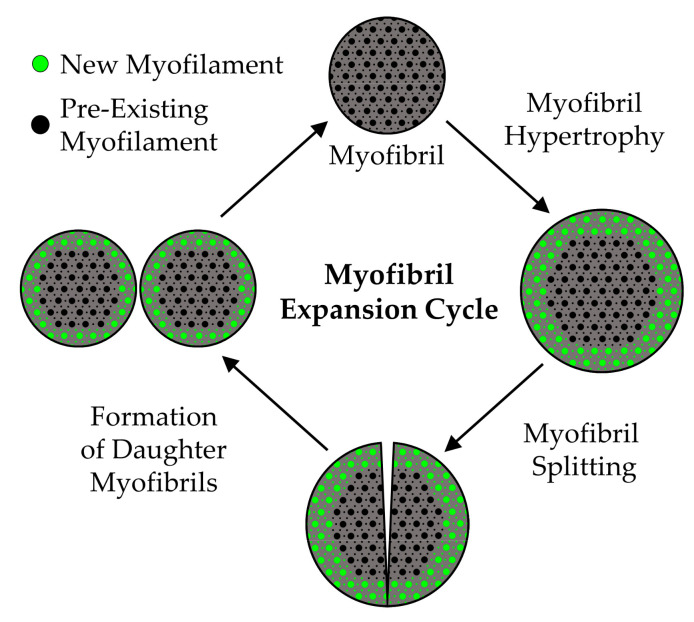
Illustration of the proposed “Myofibril Expansion Cycle”.

**Table 1 cells-09-01658-t001:** The Structural Adaptations that Drive the Mechanical Load-Induced Growth of Skeletal Muscle.

Adaptation	Evidence	Gaps in Knowledge
**Longitudinal Growth of Fascicles**	High	Does mechanical loading alter the number of fascicles? Can mechanical loading lead to the addition of new myofibers in-series?
**Radial Growth of Fascicles**	High	To what extent does myofiber hyperplasia, myofiber splitting, and the lengthening of myofibers with intrafascicular terminations contribute to the radial growth of fascicles?
**Myofiber Splitting**	Low	Do physiologically relevant models of mechanical loading induce myofiber splitting?
**Myofiber Hyperplasia**	Low & Controversial	To what extent, if any, does myofiber hyperplasia contribute to the radial growth of fascicles?
**Longitudinal Growth of Myofibers**	Mixed - Model Dependent	Do physiologically relevant forms of mechanical loading induce the longitudinal growth of myofibers? Where, and how, are new sarcomeres added during the longitudinal growth of myofibers?
**Radial Growth of Myofibers**	Extremely High	Is mechanical load-induced myofiber hypertrophy driven by myofibril hypertrophy and/or myofibril hyperplasia?
**Sarcoplasmic Hypertrophy**	Low & Controversial	Are there specific conditions during which sarcoplasmic hypertrophy might make substantive contribution to the mechanical load-induced growth of myofibers?
**Myofibril Hypertrophy**	Low	Does mechanical loading lead to myofibril hypertrophy? Where are new myofilaments deposited during myofibril hypertrophy?
**Myofibril Hyperplasia**	Very Low	Does mechanical loading lead to myofibril hyperplasia? Are new myofibrils generated via the process of myofibril splitting?

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
