# Peer review of "Identifying the Structural Adaptations that Drive the Mechanical Load-Induced Growth of Skeletal Muscle: A Scoping Review"

_cells, 2020, doi:10.3390/cells9071658_

Round 1

Reviewer 1 Report

Thank you for giving me a chance to review this paper. This is very well-written review about mechanical load-induced growth of skeletal muscle. The literature review is adequate and their figures are well illustrated for readers to understand about the skeletal muscle physiology. I do not find any problems in this paper. Congrats.

Author Response

Thank you for giving me a chance to review this paper. This is very well-written review about mechanical load-induced growth of skeletal muscle. The literature review is adequate and their figures are well illustrated for readers to understand about the skeletal muscle physiology. I do not find any problems in this paper. Congrats.

Thank you for the kind and favorable comments.

Reviewer 2 Report

Dear authors,

I would like to congratulate you to this comprehensive review on the structural adaptations in mechanical load-induced muscle growth.

I enjoyed reading your manuscript and most of the questions that were arising while reading it were answered later on.

However, I have some comments, that might be worth considering:

  • On page 5, line 157, you stated that regional differences in the magnitude of radial growth are an overlooked aspect… Why is it important? What can be the consequences of and for the application of exercise training?
  • Page 7, line 229ff and in general; what are the (different?) consequences of radial vs longitudinal growth regarding strength development and muscle function, if there are any? Can you speculate on specific training methods to favour one adaptation upon the other?
  • What is the role of satellite cells in the response to mechanical load? On which level of adaptation do they work?...
  • What is the physiological reason, regarding structural adaptations, why weightlifters/powerlifters, despite being stronger, do show lower muscle size than bodybuilders.
  • Which resistance training methods would you recommend, and why, to maximise mechanical load-induced muscle growth?
  • Within the same context – I would appreciate adding recommendations regarding hypertrophy training (vs strength training) to the take home message section.
  • Finally, some remarks in the final section of the manuscript regarding muscle mass, health and aging, related to the first paragraph of your Introduction would round up the Story.

Author Response

Dear authors,

I would like to congratulate you to this comprehensive review on the structural adaptations in mechanical load-induced muscle growth. I enjoyed reading your manuscript and most of the questions that were arising while reading it were answered later on.

  • Thank you for the positive remarks, we are glad that you enjoyed reading the manuscript.

Some comments, that might be worth considering:

  1. On page 5, line 157, you stated that regional differences in the magnitude of radial growth are an overlooked aspect… Why is it important? What can be the consequences of and for the application of exercise training?
  • We tried to highlight the importance of this point in the body of the paragraph that preceded the statement. Specifically, the paragraph describes how regional differences in the magnitude of radial growth can explain why the increase in CSA reported in studies is often greater than the increase that is observed for muscle volume/mass. We focused on this point to ensure that the reader appreciates that growth does not always occur uniformly throughout the muscle.

  1. Page 7, line 229ff and in general; what are the (different?) consequences of radial vs longitudinal growth regarding strength development and muscle function, if there are any?
  • Theoretically it can be argued that longitudinal growth will lead to faster contraction speed whereas radial growth will lead to increased force production. However, numerous variables could impact this general outcome and we are not aware of any studies that have directly addressed the topic.

  1. What is the role of satellite cells in the response to mechanical load? On which level of adaptation do they work?...
  • Whether satellite cells play a role on mechanical load-induced growth is a highly debated topic and we believe that addressing this topic would go well beyond the intended scope of our review. To appreciate the current debates that surround this topic please refer to the following references [1, 2].

  1. What is the physiological reason, regarding structural adaptations, why weightlifters/powerlifters, despite being stronger, do show lower muscle size than bodybuilders.
  • This is an interesting question. We would, however, dispute the notion that weightlifters/powerlifters, of the same bodyweight, have lower muscle size than comparable bodybuilders. Bodybuilders traditionally have lower body fat and so their musculature is simply more visible. In contrast, since weightlifting/powerlifting is not esthetically oriented their muscles are not as visible, but in the few comparisons of muscle fibers of bodybuilders versus powerlifters (cited in our paper) we note that muscle fibers are not different in size. A confounding issue is the use of anabolic androgenic steroids in bodybuilders. An interesting recent study shows, quite convincingly, that longer-term resistance-trained persons – regardless of their training mode – were stronger due to their greater muscle cross-sectional area [3]. That bodybuilders may not be as strong in practice is, therefore, most likely due to their not practicing lifting heavier weights (i.e., a neural effect). We note that such neural effects can occur even in experienced lifters, that is they are confined as most conventional thinking would surmise, so there is always ‘room’ for improvements in strength [4].

  1. Which resistance training methods would you recommend, and why, to maximise mechanical load-induced muscle growth? Within the same context – I would appreciate adding recommendations regarding hypertrophy training (vs strength training) to the take home message section.
  • The reviewer raises a very interesting and yet complex question. We could easily write a whole new review paper on this very topic, but instead refer to reviewer to several recent papers that have attempted to answer this question [5-8].

  1. Finally, some remarks in the final section of the manuscript regarding muscle mass, health and aging, related to the first paragraph of your Introduction would round up the Story.
  • Thank you for this comment, the final section has been revised accordingly.

References

  1. Murach, K.A., C.S. Fry, T.J. Kirby, J.R. Jackson, J.D. Lee, S.H. White, E.E. Dupont-Versteegden, J.J. McCarthy, and C.A. Peterson, Starring or Supporting Role? Satellite Cells and Skeletal Muscle Fiber Size Regulation. Physiology (Bethesda), 2018. 33(1): p. 26-38.
  2. Cornelison, D., "Known Unknowns": Current Questions in Muscle Satellite Cell Biology. Curr Top Dev Biol, 2018. 126: p. 205-233.
  3. Maden-Wilkinson, T.M., T.G. Balshaw, G.J. Massey, and J.P. Folland, What makes long-term resistance-trained individuals so strong? A comparison of skeletal muscle morphology, architecture, and joint mechanics. J Appl Physiol (1985), 2020. 128(4): p. 1000-1011.
  4. Balshaw, T.G., G.J. Massey, T.M. Maden-Wilkinson, M.B. Lanza, and J.P. Folland, Neural adaptations after 4 years vs 12 weeks of resistance training vs untrained. Scand J Med Sci Sports, 2019. 29(3): p. 348-359.
  5. Schoenfeld, B.J., J. Grgic, D. Ogborn, and J.W. Krieger, Strength and Hypertrophy Adaptations Between Low- vs. High-Load Resistance Training: A Systematic Review and Meta-analysis. J Strength Cond Res, 2017. 31(12): p. 3508-3523.
  6. Schoenfeld, B.J., D. Ogborn, and J.W. Krieger, Effects of Resistance Training Frequency on Measures of Muscle Hypertrophy: A Systematic Review and Meta-Analysis. Sports Med, 2016. 46(11): p. 1689-1697.
  7. Grgic, J., B.J. Schoenfeld, T.B. Davies, B. Lazinica, J.W. Krieger, and Z. Pedisic, Effect of Resistance Training Frequency on Gains in Muscular Strength: A Systematic Review and Meta-Analysis. Sports Med, 2018. 48(5): p. 1207-1220.
  8. Morton R.W., L. Colenso-Semple, and S.M. Phillips. Training for strength and hypertrophy: an evidence-based approach. Curr. Opin. Physiol. 2019. 10: 90-95, 2019